# The Prognostic Value of (1→3)-β-D-Glucan in COVID-19 Patients with and Without Secondary Fungal Disease

**DOI:** 10.3390/jof11090656

**Published:** 2025-09-05

**Authors:** Udari Welagedara, Jessica Price, Raquel Posso, Matt Backx, P. Lewis White

**Affiliations:** 1Public Health Wales Mycology Reference Laboratory, UHW, Cardiff CF14 4XW, UK; 2Centre for Trials Research, Division of Infection and Immunity, Cardiff University, UHW, Cardiff CF14 4YS, UK

**Keywords:** (1→3)-β-D-Glucan, mortality, infections, antifungals, invasive, fungal disease

## Abstract

Background: The presence of (1→3)-*β*-D-Glucan (BDG) in serum may be indicative of invasive fungal disease (IFD), but even without IFD, elevated BDG can be associated with adverse patient outcomes. Methods: COVID-19-infected patients (n = 125) who were screened for IFD with fungal biomarkers were evaluated to assess the prognostic value of BDG. BDG was correlated with patients’ mortality, considering the influences of IFD and anti-fungal therapy (AFT). Results: A BDG concentration > 31 pg/mL was associated with significant mortality in the absence of documented IFD and without subsequent antifungal therapy (≤31 pg/mL: 28% vs. >31 pg/mL: 91%; *p* = 0.0001). In patients without IFD but with BDG > 31 pg/mL, mortality dropped to 50% when AFT was administered. In patients with BDG > 31 pg/mL and neither IFD nor AFT, the average probability of death was 3.38-fold greater. Conclusions: Elevated serum BDG is associated with significant mortality in COVID-19-infected patients without IFD, irrespective of AFT. A BDG-associated proinflammatory response might be driving the high mortality. BDG serves as a prognostic marker in COVID-19-infected patients with or without IFD. When BDG is very low (≤31 pg/mL) the likelihood of death remains consistent with the background mortality rates for COVID-19 within the ICU.

## 1. Introduction

COVID-19 can give rise to a wide spectrum of disease manifestations from asymptomatic infection to life-threatening multiple organ failure [1]. There is substantial risk of secondary invasive fungal disease (IFD) during COVID-19 infection due to immunosuppressive medications and virus-related immune dysregulation [2]. IFD can lead to deleterious effects in COVID-19-infected patients if not diagnosed and treated promptly. A considerable rise in mortality (48.5%) was observed in patients with secondary IFD during the COVID-19 pandemic compared to baseline rates [2].

Determining a prognostic marker in COVID-19 is invaluable due to its unpredictable clinical course. Different blood-based biomarkers such as PCT, CD4+/CD8+ ratio, IL-6, etc., have been evaluated in previous studies showing correlation with disease severity [1,3]. (1–3)-beta-D-glucan (BDG) is a polysaccharide in most pathogenic fungal cell walls, including *Candida* and *Aspergillus* species, and is considered a broad fungal marker (excluding most cryptococci and zygomycetes) that is useful in early detection of IFD [4]. Apart from its presence in fungi, BDG is considered a common component in the human diet, as well as in some bacteria, cereals (barley and oats) and seaweeds [5]. Therefore, interpretation of BDG may be challenging due to false positives associated with translocation, clinical conditions and interventions [6].

Serial measurement of BDG has shown good prognostic value in ICU patients with proven invasive candidiasis (IC), where a decrease in BDG was associated with reduced mortality [7]. While BDG is commonly used to screen for IFD in susceptible patients, using its negative predictive value to exclude disease, new evidence is emerging indicating that it generates profound proinflammatory responses with the involvement of TH-1 and TH-17 cells and is associated with poor prognosis, irrespective of the presence of IFD [8,9].

As the majority of IFD in COVID-19-infected patients in the Western Hemisphere is caused by *Aspergillus* (21.5%) and *Candida* (21.5%) species [2], evaluating the value of BDG in COVID-19-infected patients may be beneficial for predicting mortality. At the start of the COVD-19 pandemic and in recognition of the likely increased risk of IFD in critical-care patients with severe respiratory viral infections, a diagnostic algorithm was introduced [10]. BDG testing (amongst other mycological tests) formed part of the diagnostic strategy for IFD screening of high-risk critical-care patients in Wales with COVID-19. The clinical performance and utility of this approach was previously described and demonstrated the significant negative impact of IFD in this cohort and the benefits of appropriate AFT [10]. Given the emerging evidence indicating BDG positivity to be a poor prognostic marker, particularly when positivity is prolonged, it was decided to evaluate the impact of BDG positivity in the previously defined cohort. The current study was designed to assess the prognostic value of BDG in COVID-19-infected patients with or without defined IFD and antifungal therapy (AFT).

## 2. Materials and Methods

### 2.1. Patients and Study Design

All patients screened for BDG in serum/plasma (125 of the original 135 patients) during the first wave of the pandemic in Wales were included in the current study. Testing was performed, and all data were retrieved as part of routine diagnostic assessment. The current evaluation was undertaken as a retrospective, anonymous evaluation with no impact on patient management, therefore not requiring ethical approval.

Cases of IFD were defined using the EORTC/MSGERC definitions or the ECMM/ISHAM CAPA guidelines [11,12]. The recovery of *Candida* spp. via blood culture was deemed definitive of candidaemia, whereas recovery from a central venous catheter was considered *Candida* line infection.

Antifungal therapy was administered as part of routine care during the COVID-19 pandemic. Subsequently, its administration was variable with some patients receiving it unnecessarily due to empirical antifungal strategies, while other did not receive AFT as a diagnosis of IFD was achieved post-mortem [10]. No information on the cause of death was available and crude mortality rates are provided.

### 2.2. Detection of (1→3)-β-D-Glucan

BDG testing of serum or plasma samples was performed using the Associates of Cape Cod Fungitell assay, following the manufacturer’s instructions. Samples were tested in duplicate and the following thresholds were used to interpret results:

Negative, ≤60 pg/mL; Indeterminate, 60–79 pg/mL; Positive, ≥80 pg/mL

### 2.3. Statistical Analysis

Receiver operator characteristic curve analysis was performed to determine an optimal threshold for correlating BDG concentration and prognosis. A value of >31 pg/mL was deemed optimal via the Youden’s index score. For analytical purposes, the impact of a higher BDG value of ≥60 pg/mL was also evaluated. Logistic regression analysis was performed to determine any association between increasing BDG concentration and prognosis. An overall mortality rate for the entire cohort was determined, from which additional mortality rates were calculated dependent on the designated BDG threshold and subsequent diagnosis of IFD/fungal infection and/or the use of appropriate AFT. AFT was considered appropriate if it correlated with guidelines for the management of a particular IFD or if it were deemed appropriate for the fungal species cultured. For proportional values, 95% confidence intervals (95% CI) were calculated, which were also used, along with Fisher’s exact test to determine the significance of any differences between proportional values. When comparing means/medians, representative *t*-tests were performed. For all statistical comparisons a *p* value of ≤0.05 was considered significant.

## 3. Results

### 3.1. Population

One hundred and twenty-five COVID-19-infected patients requiring critical care management were included in this study (Table 1). There was no association between the presence of bacteraemia and raised BDG concentrations. Ninety-one percent of patients received at least one antibacterial agent; while the use of antibacterial agents was associated with raised BDG concentrations, this association was deemed insignificant when accounting for the influence of IFD (Table 1). Generally, patient co-morbidities were comparable, although patients with renal conditions were proportionally greater in the higher BDG population.

The overall mortality was 38% (47/125; 95% CI: 29.6–46.3). The male/female ratio was 2.29/1, with an average age of 58 and 54 years for men and women, respectively. There was a significant correlation between increasing age and mortality (odds ratio: 1.0638 (95% CI: 1.0271–1.1018; *p*: 0.0006)) (Figure 1). There was no linear correlation between age and BDG concentration (odds ratio: 1.0046 (95% CI: 0.9728–1.0374; *p*: 0.7802)). Mortality rate in men was 41% (36/87, 95% CI: 32–52), compared to 29% (11/38, 95% CI: 17–45) in women (*p*: 0.2303). There was no significant correlation between gender and mortality (odds ratio: 1.7326 (95% CI: 0.7626–3.9366; *p*: 0.1819)).

### 3.2. (1→3)-β-D-Glucan Concentration and Mortality

In 89/125 patients, the highest recorded BDG concentration was ≤31 pg/mL (81% (63/78) of surviving patients compared to 55% (26/47) of deceased patients). Subsequently, the mortality rate was significantly greater when BDG concentrations were >31 pg/mL (58%) compared to ≤31 pg/mL (29%) (Table 2, Figure 1 and Figure 2).

Twenty-eight percent of patients who died (13/47) had a BDG concentration > 60 pg/mL compared to 15% of patients who survived (12/78) (*p*: 0.1103). Nineteen percent of patients who died (9/47) had a BDG concentration > 80 pg/mL compared to 14% of patients who survived (11/78) (*p*: 0.4617) and 11 percent of patients who died (5/47) had a BDG concentration > 500 pg/mL compared to 8% of patients who survived (6/78) (*p*: 0.7459). Subsequently, there was no significant linear correlation between increasing BDG concentration and mortality.

Logistic regression analysis confirmed a significant correlation between a BDG concentration > 31 pg/mL and mortality (odds ratio: 3.3923 (95% CI: 1.5169–7.5863; *p*: 0.0026)). Multivariable logistic regression analysis confirmed both increasing age and a BDG concentration > 31 pg/mL to be significantly associated with mortality (overall model fit *p*: < 0.0001), where across the age range, the presence of a BDG concentration > 31 pg/mL increased the probability of mortality by a mean of 2.35-fold (Figure 1).

There was a trend between a BDG concentration > 60 pg/mL and mortality (odds ratio: 2.1029 (95% CI: 0.8662–5.1057; *p*: 0.1008)) and the mortality of patients with a maximum BDG concentration above this threshold was, on average, 1.73-fold greater (Figure 1).

### 3.3. Antifungal Therapy and Mortality

Thirty-four percent (42/125) of patients received AFT deemed appropriate for the management of their IFD. When not considering BDG concentration, the mortality rate when AFT was administered was comparable to that in the absence of AFT (16/42: 38.1% vs. 31/83: 37.4%, *p*: 1.0000). Mortality remained similar for patients receiving AFT, regardless of the calculated BDG concentration (Table 2). However, in patients not receiving AFT, the presence of a BDG concentration > 31 pg/mL was associated with a significant increase in mortality (92%) compared to those with BDG ≤ 31 pg/mL (27%). Using a higher BDG threshold of ≥60 pg/mL was associated with a 100% mortality rate in patients not receiving AFT, significantly greater than in patients not receiving AFT but with a BDG concentration < 60 pg/mL (33%, *p*: 0.0008).

### 3.4. Mortality Associated with Invasive Fungal Disease

Most patients (n = 99) did not have IFD diagnosed (mean BDG concentration: 56 pg/mL), with 26 patients diagnosed with IFD (mean BDG concentration: 212 pg/mL), including 10 CAPA cases (5 probable and 5 possible (mean BDG concentration: 259 pg/mL)), 12 IC (8 *Candida* line infections (mean BDG concentration: 96 pg/mL), four candidaemia (mean BDG concentration: 273 pg/mL) and two *Candida* peritonitis (mean BDG concentration: >500 pg/mL)) and two cases of fungaemia (mean BDG concentration: 41 pg/mL) (Table 1).

When not considering the BDG concentration, the mortality rate when IFD/line infection was diagnosed was comparable to that in the absence of IFD (11/26: 42.3% vs. 36/99: 36.3%, *p*: 0.6511). Mortality remained similar for patients with IFD, regardless of the calculated BDG concentration (Table 2). However, in patients without IFD, the presence of a BDG concentration > 31 pg/mL was associated with a significant increase in mortality (74%) compared to those with BDG ≤ 31 pg/mL (28%). This was maintained when using a higher BDG threshold of ≥60 pg/mL, the mortality rate being 83% in patients without IFD compared to 30% in patients without and with a BDG concentration < 60 pg/mL (53%, *p*: 0.0006).

### 3.5. Mortality Associated with Invasive Fungal Disease and the Use of Antifungal Therapy

Eighty-eight percent (23/26) of patients with IFD/fungal line infection received appropriate AFT. The mortality rate in patients with IFD/fungal line infection receiving appropriate AFT, not considering the BDG concentration, was 39% (9/23) compared to 67% (2/3) of patients with IFD/fungal line infection who did not receive AFT (*p*: 0.5558). There was no difference in mortality in patients with IFD/fungal line infection receiving appropriate AFT, regardless of the calculated BDG concentration (Table 2). In patients with IFD/fungal line infection not receiving appropriate AFT, 100% died when the BDG concentration was >31 pg/mL, compared to 0% when the BDG concentration was ≤31 pg/mL, although this did not reach statistical significance due to the limited number of patients (Table 2).

Nineteen percent (19/99) of patients without IFD/fungal line infection received AFT. The mortality rate in patients without IFD/fungal line infection receiving AFT when not considering the BDG concentration was 37% (7/19) compared to 36% (29/80) of patients without IFD/fungal line infection who did not receive AFT (*p*: 1.000). There was no difference in mortality in patients without IFD/fungal line infection receiving AFT, regardless of the calculated BDG concentration (Table 2). In patients without IFD/fungal line infection not receiving AFT, 91% died when the BDG concentration was >31 pg/mL, compared to 28% when the BDG concentration was ≤31 pg/mL (*p*: 0.0001) (Table 2).

Multivariable logistic regression analysis confirmed that age and a combined variable where BDG was >31 pg/mL, but there was no IFD or AFT, were significantly associated with mortality (*p* < 0.0001). While the association with age generated a significant, yet minimal odds ratio (1.0472, 95% CI: 1.0111–1.0846, *p*: 0.01), the combined variable generated a significant and substantial odds ratio (25.7889, 95% CI: 3.0357–219.0793, *p*: 0.0029) that equated to, on average, a substantial 3.38-fold increase in the probability of death across all ages (Figure 1).

## 4. Discussion

COVID-19 infection can be complicated by IFD leading to increased mortality. Strategic mycological testing plays a major role directing appropriate antifungal therapy and survival of these patients [10]. BDG is a major cell wall component of many pathogenic fungi, aiding the diagnosis of IFD through its detection in serum using validated and reproducible techniques [13]. Its high negative predictive value has made it suitable as a surrogate marker excluding IFD, stratifying antifungal treatment and the duration of empirical AFT [7,14,15].

BDG is a pathogen-associated molecular pattern (PAMP) molecule recognized by various immune receptors of immune cells such as monocytes, macrophages, natural killer cells and dendritic cells. C-type lectin receptors, mainly dectin-1, play a major role in the proinflammatory immune response associated with BDG with the involvement of T-helper 1 and T-helper 17 cells [8,9]. In sepsis, bacterial and fungal PAMPs can be translocated from the gut, including endotoxins (LPS) from Gram-negative bacteria and BDG from fungi or other non-fungal sources causing a severe inflammatory response [16].

The prognostic value of high BDG concentrations leading to a worse clinical outcome, likely due to a proinflammatory host innate immune response, is an emerging concern [4] and studies have evaluated the value of BDG as a prognostic marker in different patient groups [7,9]. In patients following abdominal surgery, significant morbidity and mortality were associated with persistent or increasing BDG in their circulation regardless of IC or antifungal treatment, highlighting its significance as a prognostic indicator [9]. The significance of BDG on the prognosis of critical care COVID-19 patients, where all-cause mortality rates are already substantial (approx. 30%) and who are susceptible to secondary IFD, is less clear [17]. While studies have investigated the prognostic implications of the presence of serum/plasma BDG during COVID-19 infection, correlation with IFD (an obvious source of BDG) and its subsequent treatment have not been thoroughly investigated.

At the start of the pandemic in March 2020, an algorithm for diagnosing IFD within critical care COVID-19 patients was introduced across Wales and this involved BDG screening of serum. Data from the first wave of the pandemic were assessed to determine the prognostic value of BDG in COVID-19-infected patients, according to the presence of documented IFD and/or subsequent AFT. Predictably, increasing age was a demographic factor that correlated with mortality in critical care patients with COVID-19 (Figure 1). This has been documented previously, where the mortality rate was significantly greater in patients ≥ 65 years old (42%) compared to the younger cohort (11%), but age was not the sole factor in predicting mortality [18]. In the current study, mortality was 30% for patients < 65 years old, rising to 58% for the older cohort. When an age threshold of 50 years was employed the mortality rate was 18% and 47%, respectively, for the younger and older cohorts.

While there was no correlation between increasing age and BDG concentration, the presence of a BDG concentration > 31 pg/mL further increased the probability of death across the age range and on average more than doubled the likelihood of death (Figure 1). The significance of this finding remained at higher BDG concentrations, more typically associated with IFD (e.g., >60 pg/mL). During COVID-19 infection, particularly post the acute phase, higher BDG concentrations have been associated with pro-inflammatory responses linked to increased levels of NF-κB signalling [19]. Given that both respiratory and gastro-intestinal inflammation are common symptoms during COVID-19 infection and increased membrane permeability has been shown to be associated with higher plasma levels of zonulin (a protein that regulates the permeability of intestinal membranes), translocation of BDG from commensal or colonizing fungi within these host environments appears to be a plausible source for BDG within the host’s circulation [20]. While it is convenient to apply existing BDG thresholds optimized for the diagnosis of IFD (>60 or >80 pg/mL) to this prognostic approach, it should be accepted that these thresholds will likely differ given the different goals. Indeed, during IFD fungal burdens and subsequent levels of BDG would typically be higher than when compared to BDG translocation sourced from commensal/colonizing fungi. As a result, it is likely that optimal BDG thresholds for defining prognosis will differ from those for diagnosing IFD, and ROC analysis was performed to identify an optimal BDG threshold for predicting prognosis. In previous studies assessing the prognostic value of BDG testing, a threshold of >40 pg/mL was deemed optimal and is comparable to the optimal threshold identified in the current study (>31 pg/mL), given the inter-assay variability of the test [19,21]. Applying a BDG threshold of >40 pg/mL to determine patient prognosis in this current study remains significant (OR: 3.1034, 95% CI: 1.3439–7.1667, *p*: 0.0073).

In patients receiving AFT, mortality was typically consistent regardless of the BDG concentration and generally comparable with the mortality rates associated with COVID-19 infection in critical care patients without secondary IFD, although patient numbers are limited for some categories (Table 2). Conversely, in the absence of AFT and/or IFD, mortality was significantly greater (at least 2.6-fold) when BDG was higher (>31 pg/mL) than when BDG was low (≤31 pg/mL), where mortality rates were typical of COVID-19 in critical care. In the final multi-logistic regression model, combining BDG > 31 pg/mL in the absence of diagnosed IFD and subsequent AFT, the odds ratio associated with mortality was substantial (25.8), increasing the probability of death by an average of 3.4-fold over age alone across the age range.

Given the complexity of diagnosing IFD, particularly in cohorts lacking established host and clinical factors, the association between BDG > 31 pg/mL and increased mortality could be associated with undiagnosed IFD rather than a non-infective BDG-driven proinflammatory response [4,15,16,17,18]. In patients with no evidence of IFD other than high serum BDG, mortality was 91% when not treated with antifungals, falling to 50% when AFT was administered (*p*: 0.1108, Table 2). Sixty-five percent (17/26) of patients with documented IFD had BDG > 31 pg/mL, compared to 19% (19/99) of patients without documented IFD (*p* < 0.0001) (Table 2). The median BDG concentration of patients with documented IFD and BDG > 31 pg/mL was 386 pg/mL, compared to 70 pg/mL in those without documented IFD and BDG > 31 pg/mL (unpaired *t*-test *p*: 0.0307) (Footnotes Table 2). Thirteen of seventeen patients (76%) with documented IFD and BDG > 31 pg/mL would have been considered positive for BDG using the Fungitell threshold of 80 pg/mL, compared to seven of 19 patients (37%) without documented IFD and BDG > 31 pg/mL (*p*: 0.0228). While indicating that, in patients with documented IFD, when BDG is present in serum it is present in higher concentrations consistent with Fungitell thresholds, the response to AFT in the absence of documented IFD implies a degree of undiagnosed IFD is present and highlights the importance of diagnostic performance for detecting IFD and limiting missed diagnoses. When BDG is present at >31 pg/mL, the difference in mortality in patients receiving AFT compared to those not receiving AFT highlights a potential trigger for initiating antifungal prophylaxis when AFT for IFD has not already been administered. In this study, only 15% (19/125) of patients would have received additional AFT alongside those with documented IFD, and 64% (80/125) would not require AFT.

Unfortunately, the diagnosis of COVID-19-associated IFD is complicated, and no single test can be considered optimal, particularly when testing blood [22]. Diagnosis of IFD in COVID-19-infected patients might have been compromised due to the limited availability of deep respiratory specimens like bronchoalveolar lavage due to high risk of transmission of infection. In addition, with testing restricted to *Aspergillus* PCR, *Candida* PCR (detecting the six most common species), *Pneumocystis* PCR and *Aspergillus* antigen testing, the detection rates of IFD could have been increased if pan-fungal PCR testing was also performed. Further limitations include the number of alternative variables that could lead to a poor patient prognosis and could both contribute to and conflict with the association of BDG and mortality. While a wide number of non-fungal sources (e.g., bacterial infections, antibacterials, IVIG, surgery) have been associated with BDG false positivity, the Fungitell assay now blocks the endotoxin response within the limulus ameobcyte lysate clotting cascade, limiting the potential for false positive BDG results in the presence of Gram-negative bacteraemia. Other sources of clinical false positivity relate to the presence of BDG not necessarily associated with IFD (e.g., BDG in antibiotics derived from *Penicillium* species or translocation of BDG across mucosal membranes). Indeed, in this study renal patients (likely on dialysis) comprised a larger proportion of the population with BDG above the designated threshold than below it (Table 1). However, the innate immune system does not differentiate between different sources of BDG and potentially initiates a pro-inflammatory in response to the presence of BDG, irrespective of its source. This manuscript investigates the potential for the presence of BDG to predict mortality and, arguably, the source of BDG is less relevant, provided the assay is not detecting/cross-reacting with molecules other than BDG. Unfortunately, our study lacked data on clinical status (e.g., disease severity score) and regularly documented clinical markers of inflammation (e.g., C-reactive protein, procalcitonin), as well as evidence of cytokines/chemokines (IL-6, TNF-α) associated with pro-inflammatory innate immune response. Due to the pressure of the pandemic, antifungal susceptibility testing was not widely performed on any fungal cultures recovered from cases of IFD; as such, the absolute appropriateness of AFT according to organism cannot be guaranteed.

## 5. Conclusions

To conclude, 29% of COVID-19 critical care patients had BDG levels > 31 pg/mL with mortality at 91% when not treated with antifungal drugs, even in the absence of documented IFD. Both undiagnosed IFD and a non-infective BDG pro-inflammatory response likely contributed to this poor prognosis. Overall, this indicates that high BDG serves as a prognostic marker in COVID-19-infected patients, probably due to the associated systemic inflammatory response irrespective of IFD. In patients with a BDG concentration ≤ 31 pg/mL, mortality is typically comparable with that for COVID-19 in critical care patients. Further studies are warranted to evaluate more evidence for BDG-associated mortality, its use as a prognostic marker and measures to be taken to minimize BDG levels in serum in critically ill patients with special attention to the potential gut translocation of fungal and non-fungal PAMPs.

## Figures and Tables

**Figure 1 jof-11-00656-f001:**
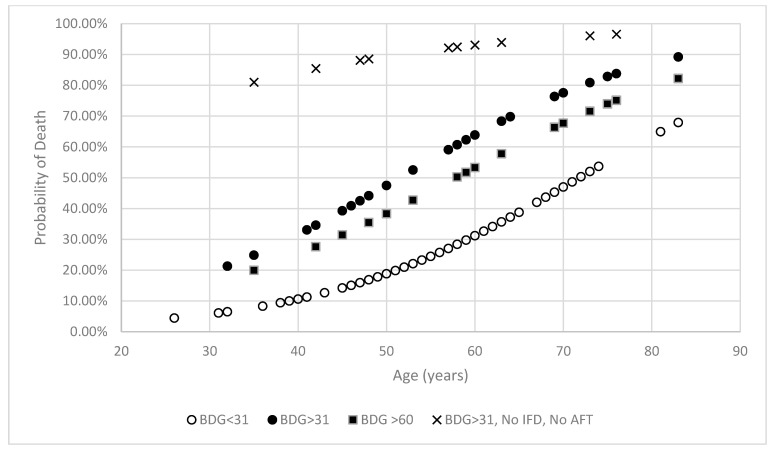
The probability of death in COVID-19 critical care patients according to age, (1–3)-β-D-Glucan (BDG) concentration and the presence of invasive fungal disease (IFD) and subsequent antifungal therapy (AFT).

**Figure 2 jof-11-00656-f002:**
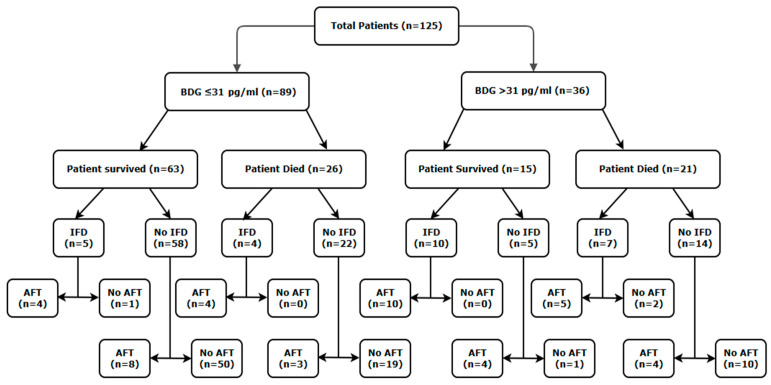
The distribution of (1→3)-β-D-glucan (BDG) concentrations among the 125 patients screened by the Associates of Cape Cod Fungitell assay, along with association with mortality with or without evidence of invasive fungal diseases (IFD) and anti-fungal treatment (AFT).

**Table 1 jof-11-00656-t001:** Basic demographics for the patients included in the study according to BDG concentration.

Parameter	Highest BDG Concentration		*p*-Value
	≤31 pg/mL (n = 89)	>31 pg/mL (n = 36)	
M/F ratio	2.6/1	1.8/1	0.3967 ^a^
Age (median)	57	58	0.8786 ^b^
Co-morbidities ^c^	HTN: 24 DM: 22 Respiratory: 21 Autoimmune: 9 Obesity: 8 Cancer: 8 Endocrine: 8 Neurological: 7 Cardiac: 6 GI: 4 Blood/vascular conditions: 3 Renal: 3 HM: 2 Other: 6 None: 18	HTN: 12 DM: 12 Respiratory: 8 Autoimmune: 5 Obesity: 3 Cancer: 2 Endocrine: 1 Neurological: 2 Cardiac: 4 GI: 5 Blood/vascular conditions: 2 Renal: 5 HM: 3 Other: 4 None: 6	0.5165 ^a^ 0.3769 ^a^ 1.0000 ^a^ 0.5426 ^a^ 1.0000 ^a^ 0.7226 ^a^ 0.4447 ^a^ 1.0000 ^a^ 0.4718 ^a^ 0.1186 ^a^ 0.6252 ^a^ **0.0435** ^a^ 0.1432 ^a^ 0.4718 ^a^ 0.8035 ^a^
IFD (Total = 26) (Organism (when available)—AFT used)	Total: 9 Fungaemia: 1 (*Rhodotorula rubra*—*Caspofungin*) CVC: 5 (All *C. albicans*—3× Fluconazole, 1× Caspofungin and 1× no AFT) Possible CAPA: 3 (All *A. fumigatus*—2× Voriconazole and 1× L-Amb)	Total: 17 Fungaemia: 1 (Unidentified yeast—fluconazole) Candidaemia: 4 (All *C. albicans*—2× Caspofungin, 1× L-Amb and 1× no AFT) CVC: 3 (1× Unidentified yeast, 1× *C. albicans* and 1× *Candida* spp.—2× Voriconazole and 1× fluconazole) *Candida* peritonitis: 2 (All *C. albicans*—1× Caspofungin and 1× fluconazole) Probable CAPA: 5 (4× *A. fumigatus* and 1× no culture—4× voriconazole and 1× L-Amb) Possible CAPA: 2 (1× *A. fumigatus* and 1× no culture—2× voriconazole)	N/A ^d^
Documented bacteraemia (Total = 58) (%)	40 (45)	18 (50)	0.6932 ^a^
Antibacterials used (all patients, Total = 114) (n, (%))	78 (79)	36 (100)	**0.0328** ^a^
Antibacterials used (excluding 26 cases of IFD (n = 9 ≤ 31 pg/mL and n = 17 > 31 pg/mL, Total = 88)) (n (%))	69 (86)	19 (100)	0.1167 ^a^

^a^ Fisher’s exact test, significant differences are shown in bold text. ^b^ Mann–Whitney test. ^c^ An individual patient may have more than one co-morbidity. ^d^ Analysis was not performed as the proportion of IFD would be expected to be greater in the higher BDG arm. Key: BDG: (1–3)-β-D-Glucan; M/F: Male/Female; HTN: Hypertension; DM: Diabetes mellitus; GI: Gastrointestinal condition; HM: Haematological condition; IFD: Invasive fungal disease; CVC: Candida line infection; CAPA: COVID-19 Associated pulmonary aspergillosis; N/A: Not applicable; AFT: Antifungal therapy; L-Amb: Liposomal amphotericin.

**Table 2 jof-11-00656-t002:** Mortality rate associated with (1→3)-β-D-Glucan concentration (Threshold: 31 pg/mL) and invasive fungal diseases with or without antifungal therapy. Significant differences are highlighted in bold text.

Population	Mortality Rate (n/N, (%))			
	BDG Concentration		Difference in Mortality (95% CI)	Significance (*p*-Value)
	≤31 pg/mL	>31 pg/mL		
Overall population (n = 125) ^a^	26/89 (29%)	**21/36 (58%) ^b^**	**29% (10.0 to 45.9)**	**0.0039**
Patients on AFT (n = 42) ^c^	7/19 (37%)	9/23 (39%) ^d^	2% (29.1 to −25.6)	1.0000
Patients not on AFT (n = 83) ^a^	19/70 (27%)	**12/13 (92%) ^e^**	**65% (37.1 to 76.2)**	**<0.0001**
Patients with IFD (n = 26) ^f^	4/9 (44%)	7/17 (41%) ^g^	3% (31.0 to −38.2)	1.0000
Patients without IFD (n = 99) ^a^	22/80 (28%)	**14/19 (74%) ^h^**	**46% (21.3 to 63.0)**	**0.0003**
Patients with IFD on AFT (n = 23) ^i^	4/8 (50%)	5/15 (33%) ^j^	17% (21.2 to −50.4)	0.6570
Patients with IFD not on AFT (n = 3) ^k^	0/1 (0%)	2/2 (100%) ^l^	100% (30.5 to −100)	0.3333
Patients without IFD on AFT (n = 19) ^a^	3/11 (27%)	4/8 (50%) ^m^	23% (18.1 to −56.2)	0.3765
Patients without IFD not on AFT (n = 80) ^a^	19/69 (28%)	**10/11 (91%) ^n^**	**63% (32.5 to 75.2)**	**0.0001**

^a^ Median BDG concentration: <31 pg/mL (range: <31 to >500 pg/mL). ^b^ Median BDG concentration: 126 pg/mL (range: 33 to 500 pg/mL). ^c^ Median BDG concentration: 39 pg/mL (range: <31 to >500 pg/mL). ^d^ Median BDG concentration: 251 pg/mL (range: 33 to 500 pg/mL). ^e^ Median BDG concentration: 78 pg/mL (range: 35 to 500 pg/mL). ^f^ Median BDG concentration: 100 pg/mL (range: <31 to 500 pg/mL). ^g^ Median BDG concentration: 386 pg/mL (range: 48 to 500 pg/mL), with 13 values >80 pg/mL (7× 500 pg/mL, 445 pg/mL, 386 pg/mL, 251 pg/mL, 156 pg/mL, 154 pg/mL and 142 pg/mL). ^h^ Median BDG concentration: 70 pg/mL (range: 33 to 500 pg/mL), with seven values >80 pg/mL (4× 500 pg/mL, 254 pg/mL, 109 pg/mL and 103 pg/mL). ^i^ Median BDG concentration: 154 pg/mL (range: <31 to 500 pg/mL). ^j^ Median BDG concentration: 445 pg/mL (range: 51 to 500 pg/mL), with 13 values > 80 pg/mL (7× 500 pg/mL, 445 pg/mL, 386 pg/mL, 251 pg/mL, 156 pg/mL, 154 pg/mL and 142 pg/mL). ^k^ Median BDG concentration: 48 pg/mL (range: 31–51 pg/mL). ^l^ Median BDG concentration: 50 pg/mL (range: 48–51 pg/mL). ^m^ Median BDG concentration: 52 pg/mL (range: 33 to 500 pg/mL), with two values >80 pg/mL (2× 500 pg/mL). ^n^ Median BDG concentration: 79 pg/mL (range: 35 to 500 pg/mL), with five values > 80 pg/mL (2× 500 pg/mL, 254 pg/mL, 109 pg/mL and 103 pg/mL). Key: BDG: (1–3)-β-D-Glucan; AFT: antifungal therapy; IFD: invasive fungal disease; 95% CI: 95% confidence interval.

## Data Availability

All data used for the calculations in this study are provided within the manuscript.

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
