# Peer review of "The Prognostic Value of (1→3)-β-D-Glucan in COVID-19 Patients with and Without Secondary Fungal Disease"

_jof, 2025, doi:10.3390/jof11090656_

Round 1

Reviewer 1 Report

The manuscript "The prognostic value of (1→3)-β-D-Glucan in COVID19 patients with and without secondary fungal diseases" reports the results of (1→3)-β-D-Glucan in COVID19 patients who were admitted to an intensive care unit in a hospital in Wales. The results are interesting and very important to publicise. 
The authors report the limitations of the study, which include the small number of patients and the fact that it was a unicentric study.

Authors need to check their references for those that are incomplete or do not comply with the journal's rules. See reference 13, lines 384-385.

Author Response

Reviewer 1

Major comments

The manuscript "The prognostic value of (1→3)-β-D-Glucan in COVID19 patients with and without secondary fungal diseases" reports the results of (1→3)-β-D-Glucan in COVID19 patients who were admitted to an intensive care unit in a hospital in Wales. The results are interesting and very important to publicise. 
The authors report the limitations of the study, which include the small number of patients and the fact that it was a unicentric study.

Detailed comments

Authors need to check their references for those that are incomplete or do not comply with the journal's rules. See reference 13, lines 384-385.

Response: Reference 13 has been updated

Reviewer 2 Report

The manuscript of Udari Welagedara and co-authors is devoted to identifying patterns of mortality and BDG concentrations in patients with Covid-19. I agree that the observations made in this article seem important, but require additional data and evidences. There is a lack of data on patients. It is unclear why 30 pg/ml was chosen as a threshold value. It is unclear why the high (>30 pg/ml) BDG rate (as the authors believe) in the group without an established IFD was registered.

Main comments

About patients. There is no general table on patient data (invasive mycosis, antifungal therapy, glucan concentration etc). It is not clear, for what purpose did patients without IFD receive antifungal therapy. On what basis was antifungal therapy prescribed (local fungal infection, such as oral candidiasis or some other reasons). And vice versa, why some patients with invasive mycosis did not receive antifungal therapy. What about other diseases, and bacterial infections. Can bacterial infection affect BDG values? And what is the cause of death.

About 30 pg/ml as prognostic value. Fungitell® Assay, which was used to determine BDG concentrations in this study, indicates that values of 10-40 pg/mL are normal for healthy individuals, and values less than 60 pg/mL are negative. The working range of concentration determination in the kit used should be specified. Specific values for patients should be given in the text (distribution of BDG values by patient groups). The article should provide clear evidences as to why the value 30 pg/ml is the threshold in the authors' opinion (now they are not there at all). Lines 284-289. Where is this data presented?

About fungal infection. The authors do not clearly indicate what causes systemic fungal infection, although it has been shown that BDG values depend on the type of pathogen, including different types of fungi Candida genus [https://doi.org/10.3390/jof11020149].

Author Response

Reviewer 2

Major comments

The manuscript of Udari Welagedara and co-authors is devoted to identifying patterns of mortality and BDG concentrations in patients with Covid-19. I agree that the observations made in this article seem important, but require additional data and evidences. There is a lack of data on patients. It is unclear why 30 pg/ml was chosen as a threshold value. It is unclear why the high (>30 pg/ml) BDG rate (as the authors believe) in the group without an established IFD was registered.

Detailed comments

Main comments

About patients. There is no general table on patient data (invasive mycosis, antifungal therapy, glucan concentration etc). It is not clear, for what purpose did patients without IFD receive antifungal therapy. On what basis was antifungal therapy prescribed (local fungal infection, such as oral candidiasis or some other reasons). And vice versa, why some patients with invasive mycosis did not receive antifungal therapy. What about other diseases, and bacterial infections. Can bacterial infection affect BDG values? And what is the cause of death.

 Response:

In respect to “There is no general table on patient data”, we have included table 1 containing basic demographics and information on IFD, bacteraemia and antibacterial use separated according to the BDG threshold.

In respect to “It is not clear, for what purpose did patients without IFD receive antifungal therapy. On what basis was antifungal therapy prescribed (local fungal infection, such as oral candidiasis or some other reasons). And vice versa, why some patients with invasive mycosis did not receive antifungal therapy”. Antifungal therapy was administered as part of routine care during the COVID-19 pandemic, subsequently its administration was variable with some patients receiving it unnecessarily due to empirical antifungal strategies while other did not receive AFT as a diagnosis of IFD was achieved post-mortem. The detail of this is covered in the original manuscript “White PL, Dhillon R, Cordey A, Hughes H, Faggian F, Soni S, Pandey M, Whitaker H, May A, Morgan M, Wise MP, Healy B, Blyth I, Price JS, Vale L, Posso R, Kronda J, Blackwood A, Rafferty H, Moffitt A, Tsitsopoulou A, Gaur S, Holmes T, Backx M. A National Strategy to Diagnose Coronavirus Disease 2019-Associated Invasive Fungal Disease in the Intensive Care Unit. Clin Infect Dis. 2021 Oct 5;73(7):e1634-e1644. doi: 10.1093/cid/ciaa1298. PMID: 32860682; PMCID: PMC7499527.” This information has been stated in lines 92-95.

In respect to what about other diseases, and bacterial infections. Can bacterial infection affect BDG values. Indeed, both bacterial infections and antibacterials can impact BDG values. While endotoxin gram-negative bacterial infections used to lead to false positive BDG results, this arm of the horseshoe crab lysate cascade is now blocked within the fungitell assay and so BDG false positivity associated with endotoxin cross reactivity is unlikely. In relation to antibacterials, penicillin sourced antibacterials can result in BDG positivity and while these are considered clinically false positive, the assay is detecting the presence of BDG that has been co-purified along with the antibacterial and so the presence of BDG within the serum represents an alternative source of BDG within the patient compared to IFD. However, the innate immune system does not differentiate between different sources of BDG and potentially initiates a pro-inflammatory in relation to the presence of BDG, irrespective of its source. This manuscript investigates the potential for the presence of BDG to predict mortality and arguably the source of BDG is irrespective, provided the assay is not detecting/cross reacting with a molecular other than BDG. Lines 401-416 provide detail on this matter, and we have also included data on these variables and any association with BDG in table 1.

In respect to cause of death, the data is crude mortality, we have no further detail. Line 96-97 now state this.

About 30 pg/ml as prognostic value. Fungitell® Assay, which was used to determine BDG concentrations in this study, indicates that values of 10-40 pg/mL are normal for healthy individuals, and values less than 60 pg/mL are negative. The working range of concentration determination in the kit used should be specified. Specific values for patients should be given in the text (distribution of BDG values by patient groups). The article should provide clear evidences as to why the value 30 pg/ml is the threshold in the authors' opinion (now they are not there at all). Lines 284-289. Where is this data presented?

 Response:

In respect to “The working range of concentration determination in the kit used should be specified”: Lines 100-104 currently state “Samples were tested in duplicate and the following thresholds were used to interpret results: Negative, ≤60 pg/ mL; Indeterminate, 60 –79 pg/mL; Positive, ≥80 pg/mL”

In respect to “Specific values for patients should be given in the text (distribution of BDG values by patient groups).” Lines 235-241 include the causes of IFD and mean BDG concentrations for each condition. Median BDG values for each of the populations where mortality rates have been calculated are included as footnotes in table 2.

In respect to “The article should provide clear evidences as to why the value 30 pg/ml is the threshold in the authors' opinion (now they are not there at all).” Lines 107-109 already state “Receiver operator characteristic curve analysis was performed to determine an optimal threshold for correlating BDG concentration and prognosis. A value of >31 pg/ml was deemed optimal”. However, we added that the optimal threshold was identified using the Youden’s score.
Lines 341-346 also read “While it is convenient to apply existing BDG thresholds optimized for the diagnosis of IFD (>60 or >80 pg/ml) to this prognostic approach, it should be accepted that these thresholds will likely differ given the different goals”. Indeed, during IFD fungal burdens and subsequent levels of BDG would typically be higher than when compared to BDG translocation sourced from commensal/colonizing fungi.” However, we supported these statements with the following sentence (lines 346-349) “Subsequently, it is likely that optimal BDG thresholds for defining prognosis will differ to that for diagnosing IFD and ROC analysis was performed in order to identify an optimal BDG threshold for predicting prognosis”

In respect to Lines 284-289. Where is this data presented? This information has been included as footnotes to table 2.

About fungal infection. The authors do not clearly indicate what causes systemic fungal infection, although it has been shown that BDG values depend on the type of pathogen, including different types of fungi Candida genus [https://doi.org/10.3390/jof11020149].

Response: We have included the causes of IFD and mean BDG concentrations for each condition in lines 235-241 and Table 1.

In my opinion, the article does not contain enough data to draw any significant conclusions.

Response:  We have included the additional information requested by the reviewer and already stated, as a limitation of the study that additional variables may have influenced the results. Nevertheless, the current model does demonstrate statistical significance.

Reviewer 3 Report

I really liked the article and believe it is very important for helping to define a marker for possible systemic fungal infection. However, I believe the results section should be revised to make it clearer for the reader.

In the Materials and Methods section, it was not clear whether the study received approval from a Human Ethics Committee. In the Results section, it would be helpful to include, for patients with confirmed fungal infections, which fungal species were identified and which antifungal treatments were administered. Additionally, information regarding antifungal susceptibility, such as whether a resistance profile was performed, could be added. These details may influence the interpretation of comparisons between patients who did or did not receive antifungal therapy.

Author Response

Reviewer 3

Major comments

I really liked the article and believe it is very important for helping to define a marker for possible systemic fungal infection. However, I believe the results section should be revised to make it clearer for the reader.

Detailed comments

In the Materials and Methods section, it was not clear whether the study received approval from a Human Ethics Committee. In the Results section, it would be helpful to include, for patients with confirmed fungal infections, which fungal species were identified and which antifungal treatments were administered. Additionally, information regarding antifungal susceptibility, such as whether a resistance profile was performed, could be added. These details may influence the interpretation of comparisons between patients who did or did not receive antifungal therapy.

Response: In relation to ethical approval, lines 84-87 currently state “Testing was performed, and all data was retrieved as part of routine diagnostic assessment. The current evaluation was undertaken as a retrospective, anonymous evaluation with no impact on patient management, subsequently not requiring ethical approval.

In relation to In the Results section, it would be helpful to include, for patients with confirmed fungal infections, which fungal species were identified and which antifungal treatments were administered, we have added the species identified and associated antifungal therapy to table 1).

In relation to information regarding antifungal susceptibility, such as whether a resistance profile was performed could be added, antifungal susceptibility testing was not widely performed during the pandemic and so this information is not available. We have added this as a limitation of the study (lines 420-423).

The table is somewhat unclear and could benefit from better organization.

To make the results more organized and easier for the reader to understand, it is essential to include clear and informative captions in the tables.

Response: Keys for all tables have been provided, and we have attempted to improve the descriptions of the different populations in table 2 (previously table 1).

Reviewer 4 Report

Review: The prognostic value of (1→3)-β-D-Glucan in COVID19 patients with and without secondary fungal diseases Major comment: The study from Welagedara and colleagues has evaluated the prognostic value of β-D-Glucan in patients with  and without secondary fungal diseases affected by COVID-19. The study provides a knowledgeable background along with well defined and described methods. Statistical analyses are in line with literature and results are clear. However there is only one major concern arising from the whole article and it is related to the study population. Authors did not present any table regarding patients’ underlying characteristics, such as comorbidities, associated diseases (both communicable and non communicable), concomitant medications of all sort. Still the work is valuable and worthy of being furtherly developed, however correlation between BDG and mortality based on the solely data presented might leak a strong pathophysiological background and explanation. Without a deep insight on cofactors and their eventual role as cofactor in the statistical analyses the research hypothesis cannot be fully demonstrated.      I would suggest to:
  1. Present patients’ population characteristics with comorbidities and medications at admission
  2. Factors affecting levels of BDG: colistin, amoxicillin/clavulanate, piperacillin/tazobactam, immunoglobulin iv., surgical operations, Candida spp colonizations, dialysis, gram negative blood stream infections. These should be included as cofactors in the multivariate logistic regression analyses as potential cofactors in the positive BDG group.
Minor comment: Materials and Methods. Line 65-72. This part should be placed at the end of the introduction section.

Author Response

Reviewer 4

Major comments

Review: The prognostic value of (1→3)-β-D-Glucan in COVID19 patients with and without secondary fungal diseases Major comment: The study from Welagedara and colleagues has evaluated the prognostic value of β-D-Glucan in patients with  and without secondary fungal diseases affected by COVID-19. The study provides a knowledgeable background along with well defined and described methods. Statistical analyses are in line with literature and results are clear. However there is only one major concern arising from the whole article and it is related to the study population. Authors did not present any table regarding patients’ underlying characteristics, such as comorbidities, associated diseases (both communicable and non communicable), concomitant medications of all sort. Still the work is valuable and worthy of being furtherly developed, however correlation between BDG and mortality based on the solely data presented might leak a strong pathophysiological background and explanation. Without a deep insight on cofactors and their eventual role as cofactor in the statistical analyses the research hypothesis cannot be fully demonstrated.   

Response: Table 1 and text Lines 127-147 have been included

Detailed comments

I would suggest to:

  1. Present patients’ population characteristics with comorbidities and medications at admission

Response: Table 1 and text Lines 127-147 have been included

  1. Factors affecting levels of BDG: colistin, amoxicillin/clavulanate, piperacillin/tazobactam, immunoglobulin iv., surgical operations, Candida spp colonizations, dialysis, gram negative blood stream infections. These should be included as cofactors in the multivariate logistic regression analyses as potential cofactors in the positive BDG group.

Response: While all of these factors can result in BDG positivity and while these are considered clinically false positive, the assay is still detecting the presence of BDG and so the presence of BDG within the serum represents an alternative source of BDG within the patient compared to IFD. However, the innate immune system does not differentiate between different sources of BDG and potentially initiates a pro-inflammatory in relation to the presence of BDG, irrespective of its source. This manuscript investigates the potential for the presence of BDG to predict mortality and arguably the source of BDG is irrespective, provided the assay is not detecting/cross reacting with a molecular other than BDG. Lines 401-416 provide detail on this matter and we have also included data on some of these variables and any association with BDG in table 1.

Minor comment: Materials and Methods. Line 65-72. This part should be placed at the end of the introduction section.

Response: We agree, this has been integrated into the introduction.

Authors did not present any table regarding patients’ underlying characteristics, such as comorbidities, associated diseases (both communicable and non communicable), concomitant medications of all sort.

Response: Table 1 included.

Round 2

Reviewer 2 Report

The authors answered all my questions and included the necessary data and clarifications in the text of the manuscript. 

The article can be accepted for publication.

Reviewer 4 Report

Authors have proprerly addressed alla major issues raised in the review

Authors have responded to all major issues detected, they have also added demographic tables required and provided adjunctive analyses of potential confounding cofactors.